# Low-Cost and Rapid Method of DNA Extraction from Scaled Fish Blood and Skin Mucus

**DOI:** 10.3390/v14040840

**Published:** 2022-04-18

**Authors:** Lang Gui, Xinyu Li, Shentao Lin, Yun Zhao, Peiyao Lin, Bingqi Wang, Rongkang Tang, Jing Guo, Yao Zu, Yan Zhou, Mingyou Li

**Affiliations:** 1Key Laboratory of Exploration and Utilization of Aquatic Genetic Resources, Ministry of Education, Shanghai Ocean University, Shanghai 201306, China; lgui@shou.edu.cn (L.G.); m210100087@st.shou.edu.cn (X.L.); 1911222@st.shou.edu.cn (S.L.); m190100017@st.shou.edu.cn (Y.Z.); 1813322@st.shou.edu.cn (P.L.); m200100094@st.shou.edu.cn (B.W.); m190100050@st.shou.edu.cn (R.T.); jguo@shou.edu.cn (J.G.); yzu@shou.edu.cn (Y.Z.); 2International Research Center for Marine Biosciences, Ministry of Science and Technology, Shanghai Ocean University, Shanghai 201306, China; 3National Demonstration Center for Experimental Fisheries Science Education, Shanghai Ocean University, Shanghai 201306, China

**Keywords:** DNA extraction, skin mucus, swabbing, viral disease diagnostic, genotyping, aquaculture

## Abstract

PCR-based DNA amplification has been one of the major methods in aquaculture research for decades, although its use outside the modern laboratory environment is limited due to the relatively complex methods and high costs. To this end, we investigated a swabbing and disc protocol for the collection of DNA samples from fish which could extract DNA from fish skin mucus by a non-invasion technique costing only $0.02 (USD) and requiring less than 30 seconds. The disc method that we chose could use the cheap filter paper to extract DNA from above 10^4^ crucian carp blood cells, which is comparable to the commercial kit. By using skin mucus swabbing and the disc method, we can obtain amplification-ready DNA from mucus to distinguish different species from our smallest fish (medaka, ~2.5 cm and crucian carp, ~7 cm) to our biggest fish (tilapia, ~15 cm). Furthermore, the viral pathogen *Carassius auratus* herpesvirus (CaHV) of crucian carp was detected using our method, which would make performing molecular diagnostic assays achievable in limited-resource settings including aquafarms and aqua stores outside the laboratory environment.

## 1. Introduction

For decades, DNA-based amplification has been one of the major methods in aquaculture research. Nucleic acid amplification in aquaculture has proven to be indispensable in laboratories for ecological studies [1] sex control breeding [2] and disease diagnostics [3]. The traditional extraction methods for nucleic acid in aquaculture are time-consuming (over 3 h), precise instrument-dependent, and require trained technicians [4]. On the other hand, common commercial rapid DNA extraction kits require only 15–30 min for DNA extraction though the comparatively high cost and requirement of laboratory devices limit the use of many DNA analyses of fish outside of the modern laboratory environment. By contrast, cellulose-based cotton paper including Flinders Technology Associates (FTA) card is biocompatible, biodegradable, and capable of adsorbing nucleic acids and has been used to obtain DNA as well as RNA from aquaculture species for molecular research in recent years [5,6]. Even common filter paper (e.g., Whatman No. 1) could extract nucleic acids from a wide range of plant, animal, and microbe samples within 30 s with convenience and low cost [7]. Several high-throughput sample-to-answer POC (point-of-care) detection of pathogens were reported based on these cellulose-based papers [6,8,9]. Obviously, cellulose-based DNA binding for molecule diagnostics is portable, inexpensive, and reliable in obtaining DNA. However, the application of this method is rarely reported in aquaculture.

The mitochondrial DNA sequences encoding the large ribosomal RNA gene are of great value for systematic and phylogenetic studies within families; sequences of the 16 S rRNA gene were always obtained for comparisons among aquatic species [1,10,11]. Crucian carp (*Carassius auratus*) is one of the major freshwater species and is a common and cheap food fish in China. To this end, we devised a simple nucleic acid extraction method from the blood of fish by using crucian carp. In recent years, significant research has been carried out using model fish due to their similarity with vertebrates as well as the improvement of aquatic research laboratory conditions, such as medaka (*Oryzias latipes*) in gonad studies [12] and zebrafish (*Danio rerio*) in genome modification and biomedical studies [13,14]. However, the majority of these fish are small bodied, and non-invasive methods of swabbing skin mucus to collect enough DNA samples have been adopted by DNA extraction kits over an in-house method which can take at least 40 min [15]. *Carassius auratus* herpesvirus (CaHV) can induce fatal infection of crucian carp and causes 100% mortality within one week, with typical signs including severe necrosis and heavy bleeding of the gills [16]. The common molecular biological extraction of such viral pathogen nucleic acids from host fish includes the collection of visceral samples and using a commercially available DNA extraction kit [17], which requires at least 1 h.

Herein, we develop a method of DNA extraction from fish blood and mucus using common and cheap filter paper (7 cm diameter, 100 pieces, USD $0.47) without specialized equipment, with results in less than 30 s, to identify different fish species and the fish pathogen CaHV. Our objective is to provide a simple, rapid, and low-cost nucleic acid testing application near aquafarm or low-resource areas.

## 2. Materials and Methods

### 2.1. Fish

Two-month-old individuals of crucian carp (*Carassius auratus*), of 7 cm average length, were purchased from a local aquafarm and gradually acclimatized to aerated water at 25 °C and fed with commercial feed (Freshwater fish extruded compound food, Xinxin Tianen Aquatic Feed company, Zhejiang Province, China) twice a day for two weeks. Three-month-old adult medakas (*Oryzias latipes*) [12,18] with a 2.5 cm average length and one-year-old adult GIFT tilapias (*Oreochromis niloticus*) [19,20] with a 15 cm average length were selected, respectively, from standard laboratory-bred stocks maintained at the Shanghai Ocean University, China, in accordance with recommendations in the Guide the Committee for Laboratory Animal Research. Diseased fish were collected from crucian carp infected with *Carassius auratus* herpesvirus (CaHV) as described before [17,21]. Briefly, healthy crucian carp were injected intraperitoneally with diseased fish tissue filtrate (viral suspension) kindly provided by Dr. Qiya Zhang (Institute of Hydrobiology, Chinese Academy of Sciences). The bleeding and infection activities were approved by Shanghai Ocean University Animal Care and Use Committee. The volume of viral suspension injected in each crucian carp was 10 μL, and PBS buffer was used as a negative control. Then, the fish were kept separately in 1 L tanks to prevent cross-contamination.

### 2.2. Blood and Mucus Samples Collection

For blood collection, each crucian carp was held under anesthesia (MS-222); the blood of the fish was collected from vessels using a 1 mL sterile syringe. The blood was immediately placed into a vacuum blood collection tube with heparin as an anticoagulant. The blood cell concentration was measured using a hemocytometer.

For mucus collection, each non-anesthetized fish was placed on a sterile dry surface and restrained firmly (Figure 1A). The mucus of these fish, which possess scales, was collected by a sterile nylon-tipped specimen collection swab (B-BSZ. BKMAM) five times along the flank of each fish, from the operculum to the caudal fin according to the previous report [15]. 

The fish used in blood collection and swabbing procedures were housed in aquaria connected to filtered, recirculating systems and monitored for 2 weeks post-sampling to check for detrimental side effects (behavioral alterations and infection/swelling at the site of DNA collection).

### 2.3. DNA Extraction Procedure by Cellulose Disc

The DNA extraction by cellulose disc process was conducted according to the previous report [7]. Discs with a 6-mm diameter were cut from filter paper using a hole puncher (Code No. 0104. Deli). Briefly, one disc was added into a 1.5 mL Eppendorf tube and sterilized before using. As shown in Figure 1B, after the sample was added to the tube with 500 μL lysis buffer (1.5 M guanidine hydrochloride, 50 mM Tris [pH 8], 100 mM NaCl, 5 mM EDTA, 1% Tween-20), nucleic acids were released and captured on filter paper by mixing gently for 10 s, allowing for the solid phase extraction of DNA in a crude sample. Then, the liquid was removed and the contaminants and amplification inhibitors were washed away for 10 s by adding 500 μL of wash buffer (10 mM Tris [pH 8.0], 0.1% Tween-20) twice. Last, 10 μL ddH_2_O was added into the tube for 2 s to dissolve DNA captured by the disc, and the 1 μL of purified DNA was transferred to the PCR reaction mix as a template.

### 2.4. Optimal Disc Determination

For the aim of determining the optimal disc among the four different types, blood samples (16 μL, 4 μL, or 1 μL of original anticoagulated blood from crucian carp) were added into tubes with a different disc and lysis buffer, respectively. Four commercially cotton filter papers including qualitative filtration paper from Sangon Biotech, Shanghai, China (Code No. F503313 medium speed) and three other types of commercial filtration paper (Code No. 101 high speed, Code No. 102 medium speed, and Code No. 103 slow speed, BKMAM, Hunan, China) were used and compared.

### 2.5. Comparison of DNA Extraction by Kit and Disc

The cell number of anticoagulated blood was counted under a microscope and then ten-fold serially diluted to the ideal concentration by sterile PBS. The efficiency of DNA extraction from fish blood of different cell numbers from 10^5^ to 10^2^ was compared by the commercial ONE-4-ALL Genomic DNA Mini-Preps Kit (Code No. B618503, BBI, New York, NY, USA) following the manufacturer’s instruction and the cellulose disc method using BKMAN No. 103 slow speed filtration paper.

### 2.6. Skin Swabbing Technique Evaluation

To evaluate the reliability of the skin swabbing technique on scaled fish, mucus DNA from 27 crucian carp was extracted and amplification of the 16s rRNA gene was performed as shown in Figure 1.

### 2.7. Effect of Different Body Size and Species on DNA Sampling

To determine the relationship between the fish body size and the quality of DNA, DNA extracted from medaka, crucian carp, and tilapia (three fish of each species) were performed amplifications; the 16s rRNA gene PCR products were sent to Sangon for sequencing.

### 2.8. Detection of Virus Infection by DNA Extract from Mucus

Diseased crucian carp were collected from the fish infected with CaHV. DNA was extracted and amplification performed with the 16s rRNA or CaHV DNA polymerase gene, respectively.

### 2.9. PCR Primers and Conditions

Nucleic acid amplification was performed by PCR. 10 μL reactions were performed using 5 μL of Premix Taq^TM^ (Code No. RR003Q, TaKaRa, Beijing, China), 0.4 μL of both forward and reverse primers (10 μmol/L), 1 μL template DNA, and 3.2 μL ddH_2_O. The universal primer of the 16S rRNA gene [11] from the fish mitochondrial genome was chosen to evaluate the quality of DNA and identify fish species. Primer CaHV-P was used for virus detection. Primers and PCR cycling parameters are shown in Table 1. PCR products were visualized on a 1.2% agarose gel.

## 3. Results and Discussions

### 3.1. Common Filter Paper Can Be Used for DNA Extraction

Extraction from nucleated blood (avian or fish blood) contains very large amounts of genomic DNA and therefore the volume of the starting material has to be scaled down. According to a previous report [7], human whole blood was diluted 1 in 5 in an extraction buffer containing proteinase K before using the cellulose disc method to extract genomic DNA. Therefore, we first chose three small volumes (16 μL, 4 μL, or 1 μL) of whole fish blood to do the DNA extraction. Our method, without proteinase K, could also save resource costs and was comparable to the commercial system in its ability to extract DNA.

To test if the common qualitative filter paper could be used to extract DNA, we compared products from two companies with prices ten times different. Our method successfully extracted and amplified crucian carp blood DNA using all the filter paper we tested. Cell lysis was achieved by diluting the blood samples (16 μL, 4 μL, or 1 μL) in a 500 μL extraction buffer; all samples from blood DNA were detectable with a clear amplification product (Figure 2). Compared to the other type of filter paper and laboratory filter paper, BKMAM slow speed filter paper gave the highest band intensity which resulted in its use of it in the remaining experiment. The amplification results from 1 μL blood had relatively brighter results, which might be due to the over-dosed DNA template obtained from 4 or 16 μL blood, or the inefficiency of lysis buffer to break cells from above 1 μL blood.

### 3.2. Evaluation of Disc Method by Comparison to Kit

The concentration of blood cells in crucian carp was 10^6^/μL. The sample of cells was 10-fold diluted to the ideal concentration (10^5^ to 10^2^/μL), and the volume of each sample was 1 μL. To validate the disc extraction method in fish, we compared it with a popular commercial Kit (Figure 3). We found that the disc method can extract amplified DNA significantly faster. As the nucleated erythrocyte existed, the manufacturers’ recommended instructions suggested the blood samples were 10 μL from fish and 100 μL from humans. Comparing the small volumes of tissue extract, the kit could purify DNA from as low as 10^3^ cells and our disc method could extract DNA from 10^4^ cells, which was diluted 100 times from pure blood. Theoretically, diploid crucian carp contains a genome size of approximately 3.5 pg of DNA [22], so 1 μL of nucleated fish blood contains 10^6^ cells which maintain around 3.5 μg DNA. Therefore, 10^4^ and 10^3^ fish blood cells maintain 35 ng and 3.5 ng DNA, respectively. The sensitivity of our disc method was comparable to the commercial system. At least 0.1 ng genomic DNA could be extracted from initial concentrations of 35 ng DNA (10^4^ fish blood cells) by using the disc method and 3.5 ng DNA (10^3^ fish blood cells) by using a commercial kit. The results showed that the kit we tested in this study could be used to extract DNA from above 10^3^ fish blood cells, cost around USD $1.58, and required more than 15 min, whereas the disc method could extract DNA from above 10^4^ fish blood cells, only cost USD $0.02 and took less than 30 s.

### 3.3. Efficacy of Swabbing and Disc Technique for Fish DNA Sampling

The swabbing and disc method consistently generated sufficient DNA for successful PCR for all the 27 skin mucus samples of crucian carp (Figure 4). Only one sample (Figure 4, No. 5) showed a low yield of PCR product in comparison with the other 26 samples. The fish skin mucus swabbing method had been used in aquaculture to detect an aquaculture-related gene [23] and pathogen [24], however, our swabbing together with the disc method is much simpler and faster than any of the available procedures.

Compared to the drawing of blood, the sampling process by swabbing was efficient and non-invasive. We, therefore, preferred the swabbing method to collect DNA samples from skin mucus. No fish used in the swabbing procedures had altered behavior and no swelling or inflammation appeared at the site of DNA collection 2 weeks post-sampling. Although the sterile nylon-tipped specimen collection swab is cheap and easy to obtain, it is not absolutely necessary as a common cotton swab could also be equally efficient and even replace the disc to collect the extracted DNA (data not shown), providing an even cheaper and accurate alternative.

### 3.4. Fish Size and Species

To examine how fish body size affected the DNA yield recovered by the swabbing and disc method, three tilapias (~15 cm) and three medakas (~2.5 cm), together with three crucian carp (~7 cm) were swabbed. Swabbing medakas of the smallest size produced comparable results to the largest tilapia (Figure 5). To our surprise, the amount of extracted DNA was not related to the size of the fish. Our results indicated that the mucus from the smallest fish (medaka, ~2.5 cm) could extract enough DNA for detection. Additionally, without the use of anesthesia, the biggest fish (tilapia, ~15 cm) was hard to restrain firmly which might have led to swabbing failure that indicated one sample from tilapia had a smaller amount of PCR product than the other fish. Therefore, the potential drawbacks of the swabbing method may include low DNA quantity or poor quality DNA as in the previous report [23]. Nonetheless, the results from sequencing showed all of the PCR products of the 16s rRNA gene could be used to identify those species of fish which determines the potential application in fish ecological studies using the swab and disc method. 

### 3.5. Fish Viral Pathogen Identification

One of the ultimate applications for the skin mucus swabbing and disc method is in molecular diagnostic assays to detect aquaculture pathogens outside the modern laboratory, replacing the current labor-intensive procedures. To test the ability of our technique to detect fish disease, we infected crucian carp with the viral pathogen CaHV. Our method successfully extracted and amplified CaHV polymerase (Figure 6) before the typical symptoms were visible, although some suspected hemorrhages appeared on the external surface of diseased fish which were easily overlooked in the early infected stage (Figure 7A–C).

The POC diagnostic is now well developed with unprecedented pace in human disease detection [18,19]. However, such a POC diagnostic has not been developed in aquaculture yet. The molecular technologies in aquaculture are not easily performed outside the modern laboratory environment, which is a major bottleneck in some field-based testing. Our cheap, accurate, and reliable diagnostic methods can play an important role in aquatic disease control and health management in aquaculture.

## 4. Conclusions

In this article, we provide a detailed swabbing and disc protocol for using blood or skin mucus for the collection of DNA samples from scaled fish which could extract DNA from fish skin mucus only costing USD $0.02 and taking less than 30 s. The simplicity and speed of this method, as well as the low cost and no requirement of specialized equipment, make testing more accessible and affordable for a wide variety of aquatic organisms and is useful for applications both inside and outside the laboratory environment.

## Figures and Tables

**Figure 1 viruses-14-00840-f001:**
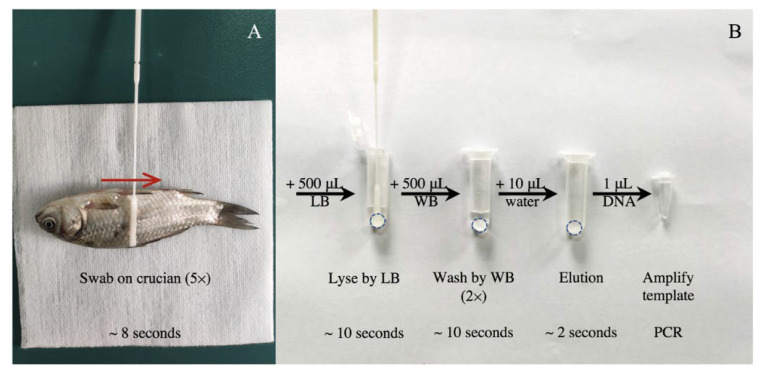
Overview of 30 s skin swabbing and disc method. (**A**) Mucus collection from a scaled fish; red arrow indicates the direction of swabbing. The process is repeated five times for about 8 s in total. (**B**) DNA is purified from the mucus. The swab with mucus is immediately dipped into a 1.5 mL Ep tube with 500 μL lysis buffer (LB) and a disc (dotted circle) for 10 s. Then, 500 μL wash buffer (WB) is transferred into the tube and incubated for 5 s two times. Last, 10 μL ddH_2_O is added into the tube to dissolve DNA captured by the disc, and the 1 μL template is transferred to the PCR reaction mix.

**Figure 2 viruses-14-00840-f002:**
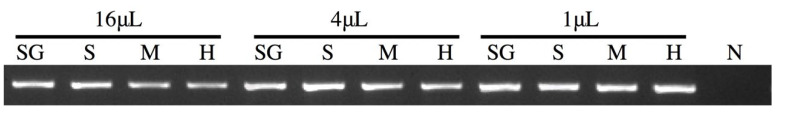
Comparison of DNA extraction efficiency of four types of filter paper under different initial blood volumes. Three different volumes (16 μL, 4 μL, 1 μL) of anticoagulant whole blood extracted from crucian carp were used for DNA extraction. Then, extracted genomic DNA was used as the template for 16s rRNA gene amplification and water was used as the negative control. SG, filter paper from Sangon Biotech; S, slow speed filtration paper from BKMAM; M, medium speed filtration paper from BKMAM; H, high speed filtration paper from BKMAM; N, negative control.

**Figure 3 viruses-14-00840-f003:**
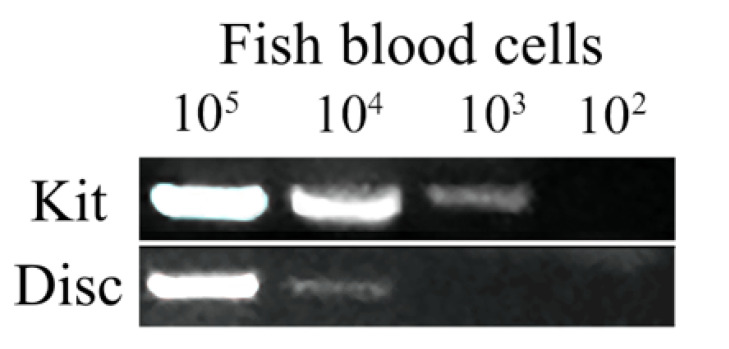
Cellulose-based paper outperforms a commercially available DNA extraction system in fish blood samples. DNA was extracted from different amounts of blood cells from crucian carp (10^5^ to 10^2^) by our disc method or ONE-4-ALL Genomic DNA Mini-Preps Kit (BBI). The eluted DNA was used in a PCR reaction with primers designed for the 16s rRNA gene.

**Figure 4 viruses-14-00840-f004:**
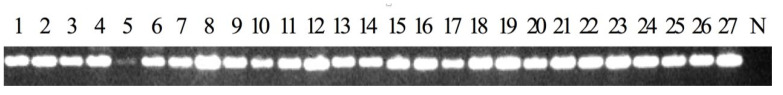
PCR results from DNA extracted from skin mucus by the swabbing and disc method. 1–27, swabs from 27 crucian carp; N, no template control. The eluted DNA was used in a PCR reaction using primers designed for the 16s rRNA gene.

**Figure 5 viruses-14-00840-f005:**
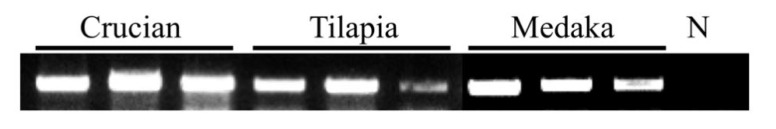
PCR results from DNA extracted from skin mucus by the swabbing and disc method from fish of different sizes: three crucian carp (~7 cm); three tilapias (~15 cm); and three medakas (~2.5 cm). N, no template control. The eluted DNA was used in PCR reaction using primers designed for the 16s rRNA gene.

**Figure 6 viruses-14-00840-f006:**
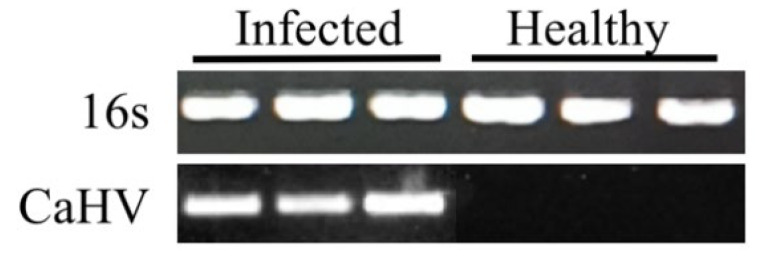
DNA extraction of diseased fish using the skin mucus swabbing and disc method. DNA was extracted from three CaHV-infected and three healthy crucian carp. The 16s rRNA gene and CaHV DNA polymerase gene were amplified, respectively.

**Figure 7 viruses-14-00840-f007:**
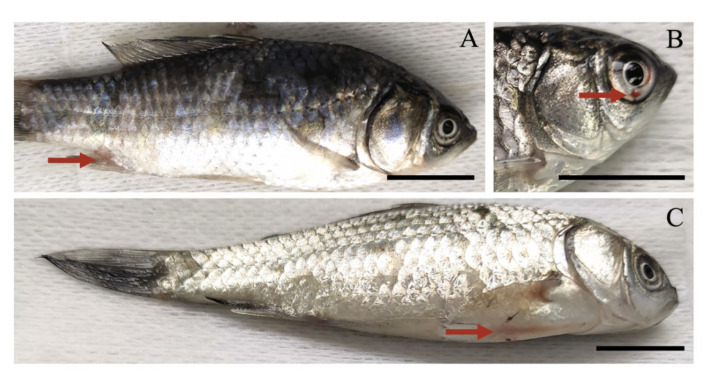
The external surface of CaHV-infected crucian carp. (**A**–**C**) Suspected hemorrhage appeared on fish, the red arrows indicate the suspected hemorrhage. The blue arrow indicates the swelling and inflammation at the blood collection site. Scale bar = 1 cm.

**Table 1 viruses-14-00840-t001:** Primers for PCR.

Names.	Sequences	Conditions	Products	Function
16 s	F: CGCCTGTTTATCAAAAACAT	94 °C for 2 min, 40 cycles of 95 °C for 30 s, 56 °C for 40 s, 72 °C for 1 min, with a final extension of 71 °C for 10 min.	~600 bp	Evaluate the quality of DNA and identify fish species [11]
	R: CCGGTCTGAACTCAGATCACGT			
CaHV-P	F: TGCTCGCTTTGATGATGGAT	94 °C for 2 min, 40 cycles of 95 °C for 30 s, 56 °C for 40 s, 72 °C for 1 min, with a final extension of 71 °C for 10 min.	328 bp	Identify CaHV infection by detection of CaHV polymerase.
	R: TTTCTTGTCTCCGGTGTCGG			

## Data Availability

Not applicable.

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
