# Peer review of "Low-Cost and Rapid Method of DNA Extraction from Scaled Fish Blood and Skin Mucus"

_viruses, 2022, doi:10.3390/v14040840_

Round 1

Reviewer 1 Report

The topic of this study is innovative and interesting, where the researchers studied the new method of DNA extraction from fish blood and mucus using the common and cheap filter paper (7cm diameter, 100 pieces, 3 RMB) without specialized equipment to identify different fish species, gender, and pathogen.

The experiments are well designed and performed. The paper is well written.

The title clearly reflects the contents of the paper

The abstract is informative

The introduction is ok and properly reviewed the relevant literature. Also, the objectives are clearly stated.

The methods used to perform the study are clear and adequate.

The results presented are adequate.

The tables and figures used to show them are adequate.

The discussion is comprehensive.

The conclusion was supported by the results and expressed the main hypothesis of the study.   However, it is suggested that authors make suggestions for future studies.

Also, there are some points that need to be corrected in the text of the article. Since the authors did not specify the number of lines in the text of the article, I put my comments in the pdf file of the article.

Author Response

We appreciate your suggestion and  valuable comments. The manuscript has now been re-written with all grammatical errors corrected according to the comments. Please see the attachment. We believe the language quality of the current version has been significantly improved.  We hope that the revised version and answers are satisfactory, and we look forward to your positive response.

Thank you again for your attentions.

Yours sincerely,

Lang Gui, Ph. D.

Reviewer 2 Report

This was a difficult manuscript to read due to the poor condition of the English language, however once that has been correct there may be useful information suitable for publication.

Most of the corrections/edits have been incorporated into the reviewed pdf of the manuscript. Some additional specific corrections/edits needed are as follows:

  1. What was the commercial source (company name and location) of the "commercial feed"?
  2. Where is the legend for Figure 1A and Figure 1B?
  3. The plural of “carp” is “carp”, not “carps”
  4. Unfortunately the fish got an “infection at the site of blood collection”. This is a significant problem with the research, suggesting that something was wrong with the author's bleeding technique, fish management/husbandry or water quality.
  5. In addition, it isn’t until the Discussion section that the reader finds out that the authors infected fish for the study (i.e. not in the Materials and Methods). Was infecting the fish approved by the Shanghai Ocean University, Committee for Laboratory Animal Research?
  6. The authors’ statement that most tilapia are sexed by DNA is not correct. Most tilapia facilities determine gender of the fish by examining external morphology, since both juvenile and adult tilapia are sexually dimorphic.
  7. Figure 6 does not add anything to support the manuscript, so it should be deleted.
  8. Currency should be either in dollars or have the equivalent of dollars in parentheses.

Author Response

We appreciate your suggestion and  valuable comments. A detailed point-by-point response to all issues is given. We hope that the revised version and answers are satisfactory, and we look forward to your positive response.

Point 1: What was the commercial source (company name and location) of the "commercial feed"?

Response 1: Thanks for the careful reading of our manuscript. The commercial feed of the fish has been added in the section 2.1

Point 2: Where is the legend for Figure 1A and Figure 1B?

Response 2: Thanks for the valuable advice. Detailed information has now been added in the legend of Figure 1: (A) Mucus collection from scaled fish, red arrow indicates the direction of swabbing. The process is repeated five times for about 8s in total. (B) DNA is purified from the mucus.

Point 3: The plural of “carp” is “carp”, not “carps”

Response 3: Thanks for the careful reading of our manuscript. We have rechecked and corrected the errors and mistakes of graphs according to the comments.

Point 4: Unfortunately the fish got an “infection at the site of blood collection”. This is a significant problem with the research, suggesting that something was wrong with the author's bleeding technique, fish management/husbandry or water quality.

Response 4: Thanks for the valuable advice. We have corrected the error. The diseased fish infected with CaHV were easily got inflammation at the blood collection site, since the virus infected fish were gaunt and in bad physical condition. Detailed information has now been added in the section 3.5 and Figure 7D.

Point 5: In addition, it isn’t until the Discussion section that the reader finds out that the authors infected fish for the study (i.e. not in the Materials and Methods). Was infecting the fish approved by the Shanghai Ocean University, Committee for Laboratory Animal Research?

Response 5: Thanks for the suggestion.  Healthy crucian carps were injected intraperitoneally with diseased fish tissue filtrate (viral suspension). The volume of viral suspension injected in each crucian carp was 10 μL, and PBS buffer was used as negative control. Then, the fish were kept separately in 1L tanks to prevent cross-contamination.  Detailed information has now been added in section 2.1.

Point 6:The authors’ statement that most tilapia are sexed by DNA is not correct. Most tilapia facilities determine gender of the fish by examining external morphology, since both juvenile and adult tilapia are sexually dimorphic.

Response 6: Thanks for the suggestion.  It is true that tilapias are sexually dimorphic, with males having a natural ability to grow faster than females. Therefore, all-male tilapias and the control of unwanted female tilapias are favored during the grow-out period. The genetic basis of sex determination in tilapia has been studied for over 50 years. Both male heterogametic XX/XY system and female heterogametic ZZ/ZW system have been identified in tilapia. Some sex-determination genes have been found in different species of tilapias. For our opinion, it is easier and cheaper to extract the DNA and identify sex-related genes of tilapia near-farm by using swab and disc, and could contribute to the development of technologies to produce all male populations. However, this part of work has no relationship to virus, so we have deleted this section according to the next comment.

Point 7: Figure 6 does not add anything to support the manuscript, so it should be deleted.

Response 7: Thanks for the valuable advice. We have rechecked and deleted the Figure 6 and the section of sex identification of fish.

Point 8: Currency should be either in dollars or have the equivalent of dollars in parentheses.

Response 8: Thanks for the careful reading of our manuscript. We have rechecked and corrected the errors according to the comments.

The manuscript has now been re-written with all grammatical errors corrected according to the comments. Please see the attachment. We believe the language quality of the current version has been significantly improved.

Thank you again for your attentions.

Yours sincerely,

Lang Gui, Ph. D.

Round 2

Reviewer 2 Report

This is still a very difficult manuscript to read due to the unacceptable English language. I have included additional English corrections that still need to be made. I still find the sampling procedure (the rough, dry handling and bleeding, not the swabbing technique) and post-sampling care inappropriate, and feel this information and associated photos should be deleted from the manuscript. The authors also have not answered whether the bleeding and infection activities were approved by the Shanghai Ocean University, Committee for Laboratory Animal Research.

Specific major corrections/edits (see underlined corrections) that are still needed:

  1. Abstract – “which could extract DNA from fish skin mucus with non-invasion by only costed United States Dollar (USD) $0.016 and lasted less than 30 seconds.” This sentence should be written as “which could extract DNA from fish skin mucus by a non-invasion technique and costing only United States Dollar (USD) $0.02 and requiring less than 30 seconds.”
  2. Introduction – “On the other hands” should be “On the other hand”.
  3. Introduction – “molecular research recent years” should be “molecular research in recent years”.
  4. Introduction - “a large number of research” should be “a large amount of research”.
  5. M&M, Fish – “crucian carps” should be “crucian carp”
  6. M&M. Blood and mucus sample collection – “fish was placed on a sterile dry surface and was restrained firmly” this type of handling procedure is the primary reason the authors had skin lesions on the fish post-sampling.
  7. M&M, Detection of virus – “DNA were extracted and performed 16s rRNA gene and CaHV DNA polymerase gene amplification, respectively.” should be written “DNA was extracted and amplification performed with the 16s rRNA or CaHV DNA polymerase gene, respectively.”
  8. Results, Common filter paper – “It might due to” should be “This might be due to”
  9. Results, Evaluation of disc – “by costed” should be “and cost”.
  10. Results, Evaluation of disc – “only costed” should be “only cost”.
  11. Results, Efficacy of swabbing – “by comparing” should be “compared to”.
  12. Results, Efficacy of swabbing – “sampling progress” should be “sampling process”.

13. Results, Fish viral pathogen – “They always got inflammation at the invasive collection site (Figure 7D) and were dead soon during the 2 weeks post-sampling” – This again is the result of the inappropriate handling of fish during sampling and suboptimal post-sampling care. This distracts from the rest of the text on diagnostic sampling and does not support the main focus of the manuscript. This information and associated photos absolutely need to be deleted.